# Development of Effective PEDV Vaccine Candidates Based on Viral Culture and Protease Activity

**DOI:** 10.3390/vaccines11050923

**Published:** 2023-04-30

**Authors:** Dae-Min Kim, Sung-Hyun Moon, Seung-Chai Kim, Ho-Seong Cho, Dongseob Tark

**Affiliations:** 1Laboratory for Infectious Disease Prevention, Korea Zoonosis Research Institute, Jeonbuk National University, Iksan 545431, Republic of Korea; daeminkk@gmail.com; 2College of Veterinary Medicine, Bio-Safety Research Institute, Jeonbuk National University, Iksan 54596, Republic of Korea; chunsu17@naver.com (S.-H.M.); leesor2@jbnu.ac.kr (S.-C.K.); hscho@jbnu.ac.kr (H.-S.C.)

**Keywords:** porcine epidemic diarrhea virus, live attenuated vaccine, pathogenicity

## Abstract

Porcine epidemic diarrhea (PED) is a highly contagious disease that has been reported annually in several Asian countries, causing significant economic losses to the swine livestock industry. Although vaccines against the porcine epidemic diarrhea virus (PEDV) are available, their efficacy remains questionable due to limitations such as viral genome mutation and insufficient intestinal mucosal immunity. Therefore, the development of a safe and effective vaccine is necessary. In this study, a virulent Korean strain of PEDV, CKT-7, was isolated from a piglet with severe diarrhea, and six different conditions were employed for serial passage of the strain in a cell culture system to generate effective live attenuated vaccine (LAV) candidates. The characteristics of these strains were analyzed in vitro and in vivo, and the CKT-7 N strain was identified as the most effective vaccine candidate, with a viral titer peak of 8.67 ± 0.29 log_10_TCID_50_/mL, and no mortality or diarrhea symptoms were observed in five-day-old piglets. These results indicate that LAV candidates can be generated through serial passage with different culture conditions and provide valuable insights into the development of a highly effective LAV against PEDV.

## 1. Introduction

Porcine epidemic diarrhea (PED) was first discovered in England in 1971 and is caused by the porcine epidemic diarrhea virus (PEDV), which was identified in 1978 [1]. Since then, PED outbreaks have occurred annually in countries such as the Republic of Korea (1992), Japan (1982), China, and Thailand (2009) [2]. In 2010, China experienced a large-scale outbreak of severe PEDV. The United States experienced a severe PED outbreak in the spring of 2013, which spread to Canada and Mexico. Outbreaks of PED have also affected countries in Asia and Europe, including the Republic of Korea, Japan, Belgium, and France, resulting in considerable losses to the livestock industry [3].

PEDV is a single-stranded positive-sense RNA virus belonging to the *Coronaviridae* family of the order *Nidovirales*. Its viral genome is approximately 28 kilobase (kb) and it consists of a 5′ untranslated region (UTR), 3′ UTR, and at least seven open reading frames (ORFs). The PEDV genome is mainly composed of RNA- encoded non-structural proteins, including replicase ORF 1a and 1b, and four structural proteins: spike (S), envelope (E), membrane (M), and nucleocapsid (N), and accessory protein ORF 3 [4]. Non-structural proteins play a crucial role in viral replication, transcription, and translation, while recent studies have shown that they also act as interferon antagonists for viral infections [5]. Among structural proteins, the S protein, which is a type I glycoprotein (GP), plays a critical role in the membrane fusion process that occurs after virus attachment to cell receptors by forming peplomers on the virion. In addition, the S protein includes major neutralizing epitopes, such as the CO-26K equivalent epitope (COE; 499–638 aa), SS2 (748–755 aa), SS6 (746–771 aa), and 2C10 (1368–1374 aa), and is associated with virus adaptation in cell culture system [6]. Investigations of genetic relationships based on sequence diversity have identified S protein as a major target for preventive strategies. The E protein aids in the viral assembly and budding process, while the highly conserved M protein is associated with E assembly. The multi-functional N protein participates in viral genome replication and virion assembly and is also involved in pathogenicity [7].

Watery diarrhea, vomiting, anorexia, and severe dehydration are symptoms of PED, which is a highly contagious enteric disease that is characterized by high morbidity and mortality in neonatal piglets [8]. The severity of the disease varies according to the age of the pigs and the strain of PEDV, which differs by time and country of origin [9]. Genetic analysis has revealed that PEDV strains can be classified into two groups: classical or recombinant, and low-pathogenic, belonging to genogroup 1 (G1); and field epidemic or pandemic, and high-pathogenic, belonging to genogroup 2 (G2). These genogroups are further subdivided into G-1a, G-1b, and G-2a, G-2b, respectively [10]. Currently, there are two main types of PEDV vaccines: live attenuated and inactivated. These vaccines have been widely used to prevent PEDV outbreaks. Live attenuated vaccines (LAVs) are generated by modifying a wild-type virus in vitro using cell culture systems and have been crucial in controlling the spread of PEDV. Several commercially available LAVs have been developed, and multiple vaccination programs have improved the survival rate of piglets, as shown by numerous vaccine studies [11]. However, despite the availability of these vaccines, the Republic of Korea has experienced PED outbreaks caused by the G2-b strain [12]. This is likely due to the genetic diversity among PEDV strains, which can limit the efficacy of current vaccines. To address this challenge, there is a need to develop more effective and safer PEDV vaccines based on the G2-b strain.

The goal of this study is to develop a LAV using virulent Korean PEDV strains (G-2b) through typical methods. The PEDV strains were initially isolated from the small intestines of infected piglets using Vero cells. The isolated PEDV strains were then serially passaged under various culture conditions, including supplementation with L-1-tosylamide-2-phenylethly chloromethyl ketone (TPCK)-treated trypsin, fetal bovine serum (FBS), and glycochenodeoxycholic acid (GCDCA) in media. These culture conditions produced six developed strains: CKT-7 T15 (with TPCK-treated trypsin), CKT-7 T15N (with TPCK-treated trypsin and GCDCA), CKT-7 N (with GCDCA), CKT-7 NF (with GCDCA and FBS), CKT-7 F (with FBS), and CKT-7 X (no supplement). This study demonstrated that all six developed strains, generated by serial passage with different culture conditions, adapted well to Vero cells and exhibited distinct biological characteristics. Moreover, the pathogenicity of the strains was evaluated in five-day-old piglets. The animal experiment showed that the groups of developed strains increased the survival rates and reduced diarrhea compared to the positive control group inoculated with a virulent wild-type parent strain. Additionally, whole-genome sequencing was conducted to analyze genetic alterations between parental and developed strains using next-generation sequencing (NGS). These results suggest that the developed strains may be suitable candidates for a novel LAV and may help prevent the spread of high-pathogenicity PEDV.

## 2. Materials and Methods

### 2.1. Cells

Vero cells were cultured in Dulbecco’s Modified Eagle Medium (DMEM; Gibco, Grand Island, NY, USA) containing 5% FBS (FBS; Gibco, Grand Island, NY, USA) and antibiotics (100 U/mL penicillin, 100 μg/mL streptomycin; Gibco, Grand Island, NY, USA) at 37 °C in a humidified 5% CO_2_ incubator.

### 2.2. Isolation of PED Virus

Porcine intestines were collected from PEDV-spreading pig farms in the Republic of Korea. The collected samples were homogenized with sea sand in DMEM, which was supplemented with antibiotics, 0.02% yeast extract (Difco, Detroit, MI, USA), 0.3% tryptose phosphate broth (Difco, Detroit, MI, USA), and 1.5 μg/mL TPCK-treated trypsin (Sigma, Chicago, IL, USA) and centrifuged at 3000 rpm for 20 min. The supernatants were filtered using a 0.22 μm syringe filter (Milipore, Burlington, MA, USA). Subsequently, the supernatants were inoculated into monolayered Vero cells and incubated for 5 days. The infected cells were passaged until distinct morphological changes were observed using microscopy (Leica microsystem, Wetzlar, Germany).

### 2.3. Live Attenuated PEDV

After isolation, the PEDV was serially passaged using six different culture conditions for a total of 180 passages, which included the addition of 1.5 μg/mL TPCK-treated trypsin, 1% FBS, or 100 μM Na-GCDCA (Na-GCDCA; Sigma, Chicago, IL, USA), either alone or in combination. The virus was diluted to 0.1 MOI and inoculated onto a monolayered Vero cell. After 1 h post-inoculation, the inoculated media was replaced with fresh virus growth media, and the cells were monitored daily for cytopathic effects (CPEs) using microscopy. The virus was stored at −80 °C once CPEs were determined to be over 80% and was used for the next passage.

### 2.4. Titration of PEDV

Virus titration was evaluated using the Reed–Muench method [13]. Briefly, Vero cells were seeded at a density of approximately 4 × 10^4^ cells per well in a 96-well plate. The virus was serially diluted at 10-fold and inoculated onto the cells, which were then incubated for 1 h. Then, the virus was removed and replaced with fresh virus growth media. The cells were observed daily using microscopy for up to 5 days post-inoculation.

### 2.5. Immunofluorescence Assay

Prior to 24 h, the Vero cells were inoculated with PEDV at 0.1 MOI. The cells were washed using phosphate-buffered saline (PBS; Gibco, Grand Island, NY, USA) and fixed with 80% cold acetone for 15 min. The fixed cells were then incubated with anti-PEDV monoclonal antibody (Median diagnostic, Chuncheon-si, Republic of Korea) for 1 h at room temperature, followed by treatment with anti-mouse secondary antibody conjugated to Alexa Fluor 488 (Cell signaling technology, Danvers, MA, USA) for 1 h at room temperature in the dark. The cells’ nuclei were stained with 4′6-diamino-2-phenylindole (DAPI; Sigma, Chicago, IL, USA), and the cells were visualized using the CELENA^®^ S digital imaging system (Logos biosystem, Anyang, Republic of Korea).

### 2.6. Evaluation of Pathogenicity

Twenty-four five-day-old piglets were obtained from a PED non-outbreak farm and randomly divided into eight experimental groups (n = 3) in separate rooms. During the acclimation period, the piglets were monitored for signs of diarrhea and overall health and were fed a replacer (Jeilfeed, Daejeon, Republic of Korea) three times per day. Subsequently, the piglets were orally inoculated with the developed strains (7.0 log_10_TCID_50_) or DMEM. In the positive control groups, the piglets were administered a virulent parent strain that was a 10% homogenized solution of intestines (passage number 0). After inoculation, clinical symptoms were monitored daily, and viral shedding in rectal swabs was measured using quantitative real-time RT-PCR. Fecal consistency scores were assigned as follows: 0 for solid, 1 for pasty, 2 for semi-liquid, 3 for liquid, and 4 for death [3]. At the conclusion of the animal experiment, the number of piglets was examined for histological lesions after autopsy. All animal experiments were approved and conducted under the Institutional Animal Care and Use Committee of Jeonbuk National University (IACUC, Protocol #: JBNU2021-078).

### 2.7. Quantitative Real-Time RT-PCR

Quantification of viral shedding from rectal swabs was analyzed using quantitative real-time RT-PCR. The rectal swab samples were collected in a clinical viral transport medium (CTM; Noble-bio, Suwon-si, Republic of Korea). The viral RNA was extracted using QIAamp viral RNA mini kit (Qiagen, Hilden, Germany) and quantified using the cellscript^TM^ RT-Q Green blue master mix (Cell-safe, Daejeon, Republic of Korea) for qRT-PCR. The RNA copy numbers were calculated using a standard curve generated with a PEDV-specific primer set based on the M gene. The primer set consisted of the forward primer PEDV-M-F (5′-GGTTCTATTCCCGTTGATGAGGT-3′) and the reverse primer PEDV-M-R (5′-AACACAAGAGGCCAAAGTATCCAT-3′) [14]. All qRT-PCR was performed using CFX96 real-time PCR (Bio-Rad laboratories, USA).

### 2.8. Histopathology and Immunochemistry

After the autopsy, the small intestines (duodenum, jejunum, and ileum) were fixed in 10% formalin and embedded in paraffin according to standard procedures. Formalin-fixed paraffin-embedded tissue blocks were sectioned into 5–8 um thicknesses using a standard rotary microtome (HM-340E; Thermo Fisher, Waltham, MA, USA) and placed on slides. The tissues were deparaffinized using xylene and treated in ethanol (100%, 90%, 70%, and 50%) for 5 min each. The tissue sections were then stained with hematoxylin and eosin (H&E) for histopathology. For immunochemistry, antigen retrieval was conducted using citrate buffer (pH 6.0) at 95 °C for 30 min and at room temperature for 20 min. Subsequently, the sections were incubated with anti-PEDV monoclonal antibody (Median diagnostic, Chuncheon-si, Republic of Korea) overnight at 4 °C. The samples were then labeled with horseradish peroxidase-conjugated anti-rabbit and anti-mouse immunoglobulin G antibodies (Vector Laboratories, Newark, CA, USA) and developed using the 3,3′-diaminobenzidine (DAB; Vector Laboratories, Newark, CA, USA) according to the manufacturer’s instructions. The samples were counterstained with methyl-green, and all slides were visualized and imaged using a light microscope (BX53, DP80; Olympus, Tokyo, Japan).

### 2.9. Next-Generation Sequencing (NGS)

Viral whole-genome sequences were determined using NGS technology. Briefly, viral RNA was extracted using a QIAamp viral RNA mini kit (Qiagen, Hilden, Germany) according to the manufacturer’s instructions, and reverse transcription was conducted using a sequence-independent single primer amplification method as described previously [15]. The final double-stranded cDNA products were quantified using a Qubit dsDNA HS assay kit (Invitrogen, Waltham, MA, USA). To generate multiplexed paired-end sequencing libraries, the double-stranded cDNA and Nextera DNA Flex Library Prep kit (Illumina, San Diego, CA, USA) were used according to the manufacturer’s instructions. The sequencing libraries were analyzed using a High Sensitivity DNA Chip on a Bioanalyzer (Agilent Technologies, Santa Clara, CA, USA), and the multiplexed libraries were sequenced using the Illumina iSeq 100 platform (2 × 150 bp).

### 2.10. Phylogenetic Tree

For the comparison of reference sequences, whole-genome sequences of PEDV strains were collected from the GenBank database. The sequence alignments were reconstructed using the Clustral W program. Phylogenetic trees based on nucleotides of the whole genome, ORF 1a, ORF 1b, S, ORF3, E, M, and N were generated using the neighbor-joining method in MEGA X software (version 11) [16]. The percentage of reliability values for each node was determined using bootstrap analysis with 1000 replicates, and the evolutionary distances were computed using the Tamura–Nei method.

## 3. Results

### 3.1. Isolation of PEDV from Porcine Intestines

Pigs from a PED outbreak farm with positive PCR results had PEDV isolated from their porcine small intestines. After three blind passages, morphological changes were observed in PEDV-infected cells, and typical CPEs, such as multinucleation, cytoplasmic fusion, and syncytium formation, were exhibited at the fourth passage. No changes were observed in the control (Figure 1A). To confirm the isolation of PEDV, an immunofluorescence assay was conducted using PEDV-specific monoclonal antibody. The PEDV-infected cells were indicated by a green signal but not in the control (Figure 1B). The newly isolated PEDV was successfully adapted in Vero cells and designated as CKT-7. Obvious CPEs were observed within 72 h post-inoculation in early passage numbers, while Obvious CPEs were observed within 20 h post-inoculation in later passage numbers.

### 3.2. Development of Live Attenuated Vaccine Candidates

To develop an effective LAV, the isolated PEDV strain was passaged in cell culture under six different conditions (Table 1). The CKT-7 strain (passage number 0) was serially passaged with TPCK-treated trypsin (1.5 μg/mL) in Vero cells up to 180 passages, which was designated as CKT-7 T15. Na-GCDCA (100 μM) was added to the virus from passage number 9, resulting in the generation of CKT-7 T15N. The concentration of TPCK-treated trypsin was gradually decreased to generate strains that could grow without TPCK-treated trypsin. The CKT-7 T10N was passaged with TPCK-treated trypsin (1.0 μg/mL) and Na-GCDCA (100 μM), while the CKT-7 T5N and CKT-7 T2N were cultivated with TPCK-treated trypsin at concentrations of 0.5 μg/mL and 0.2 μg/mL, respectively. Finally, a strain was generated with only Na-GCDCA at passage number 61 and designated as CKT-7 N. Furthermore, the strain was passaged with the addition of FBS and designated as CKT-7 NF. CKT-7 F and X strains were generated by passaging CKT-7 NF with only FBS or without any additive. Consequently, six strains were developed under six different culture conditions up to passage number 180.

### 3.3. Growth Characteristics of CKT-7 Strains

For analysis of biological characteristics, the developed strains were observed using microscopy. The CPE induced by PEDV infection was detected within 3 days post-inoculation. Strains passaged with TPCK-treated trypsin (CKT-7 T15 and T15N) were confirmed to exhibit typical CPEs, such as syncytium formation (Figure 2A). At early passage numbers, the CPE was characterized by small-scale syncytium formation, which increased in scale and developed quickly with subsequent passages. Immunofluorescence assay showed that the green signal was only observed in the syncytium region, whereas no signal was detected in the control (Figure 2B). On the contrary, other strains passaged without TPCK-treated trypsin (CKT-7 N, NF, F, and X) exhibited atypical CPEs such as cell detachment, cell lysis, and cell death (Figure 2A). The CPE of CKT-7 NF and X strains included syncytium and cell lysis, but over time, their CPE characteristics shifted to only cell lysis and detachment. In addition, CKT-7 N and F strains were only progressed by cell lysis. Interestingly, the strains grown without TPCK-treated trypsin exhibited a green signal in each cell’s cytoplasm in the immunofluorescence assay (Figure 2B). As a result, strains passaged under different culture conditions were exhibited distinguishably.

### 3.4. Growth Kinetics of CKT-7 Strains

The CKT-7 strain was initially isolated using only TPCK-treated trypsin and subsequently passaged with Na-GCDCA and/or FBS. Titration was used to measure the growth kinetics of the virus after every 10 passages (Table 2). At passage 10, the titer of the CKT-7 T15 strain was measured at 3.50 ± 0.00 log_10_TCID_50_/mL. The titer was increased with each passage and maintained above 7.00 ± 0.00 log_10_TCID_50_/mL from passage 140. The highest titer of the CKT-7 T15 strain was reached at passage 150, with a peak of 8.30 ± 0.14 log_10_TCID_50_/mL. Similarly, the titer of the CKT-7 T15N strain gradually increased after passage 10, reaching a peak titer of 8.50 ± 0.00 log_10_TCID_50_/mL at passage 150, and remained above 7.00 ± 0.00 log_10_TCID_50_/mL in subsequent passages. In contrast, the CKT-7 N strain grown without TPCK-treated trypsin had a titer of 7.17 ± 0.14 log_10_TCID_50_/mL at passage 80, which steadily increased. From passage 130, the titer was maintained above 8.00 ± 0.00 log_10_TCID_50_/mL, with the highest titer recorded at passage 140, at 8.67 ± 0.29 log_10_TCID_50_/mL. The CKT-7 NF strain exhibited titers ranging from 5.92 ± 0.14 to 8.33 ± 0.14 log_10_TCID_50_/mL, with the highest titer generated at passage 160. The CKT-7 F and X strains had relatively low titers, with the highest titer of both strains confirmed at 7.92 ± 0.14 and 7.83 ± 0.14 log_10_TCID_50_/mL, respectively. Therefore, the CKT-7 strains were adapted to each culture condition, and the CKT-7 N strain exhibited the highest titer.

### 3.5. Evaluation for Pathogenicity of CKT-7 Strains

After analyzing the biological characteristics, the pathogenicity of CKT-7 strains was evaluated in five-day-old piglets, as summarized in Table 3. Three piglets were randomly assigned to each group, and during acclimation, all piglets had normal, healthy fecal consistency. Subsequently, depending on the group, the developed strains or medium were administered orally to all piglets. The piglets were monitored daily for clinical symptoms. Rectal swabs were collected to determine viral shedding. In this animal experiment, the mortality rate and severe diarrhea rate ranged from 0% to 100%, and clinical symptoms included mild and no diarrhea, with the exception of the positive control group, which exhibited severe diarrhea within 24 h post-inoculation. Furthermore, three piglets in the positive group died within four days after inoculation (100% mortality rate). The virus shedding in rectal swabs was started at 1 dpi, and the clinical symptoms persisted until death (Figure 3C–E). The small intestines of the positive group were filled with yellowish fluids, which became thin and transparent (Figure 3B). The CKT-7 T15 and T15N groups had no mortality or severe diarrhea rates, but mild diarrhea and limited shedding were observed in the CKT-7 T15 group (Figure 3C–E). In the CKT-7 T15N group, viral shedding started at 4 dpi and the highest viral copy number was measured at 2.48 ± 0.13 log_10_copies/μL at 5 dpi (Figure 3E). Likewise, although the CKT-7 X group did not have confirmed mortality and severe diarrhea, the virus shedding in rectal swab started at 2 dpi, and the peak viral load was observed at 2.68 ± 0.49 log_10_copies/μL at 3 dpi (Figure 3C–E). The CKT-7 NF group progressed with mild diarrhea, but no virus shedding was found in the rectal swab. The small intestines of the CKT-7 T15, T15N, NF, and X groups were mildly thin and full of semi-liquid contents. In contrast, no death or clinical symptoms were observed in CKT-7 N and F groups (Figure 3A,B). Additionally, no viral genome was detected in the rectal swabs of both groups, and their small intestines were normal (Figure 3C–E). In conclusion, the pathogenicity of CKT-7 strains was evaluated in vivo in five-day-old piglets. Although there were differences in attenuated degrees, all strains were attenuated compared to the positive control, as there were no mortality or severe diarrhea rates observed. Especially the CKT-7 N and F strains were completely attenuated in neonatal piglets.

### 3.6. Analysis of Histopathology and Immunohistochemistry (IHC)

To evaluate pathogenicity, the small intestine was generally divided into three parts, duodenum, jejunum, and ileum. The tissues were stained with H&E, and IHC was used to examine for antigens using a PEDV-specific monoclonal antibody. The H&E staining revealed that the small intestines from the positive control group were characterized by viral enteritis with severe atrophy, shortened villi, and fusion. Conversely, the tissues were normal in the negative control group (Figure 4A). In the CKT-7 T15, T15N, and NF groups, the severity of small intestines was observed with moderated villi atrophy and shortening. Although the small intestines were injured, the intestinal villous atrophy was relatively mild in CKT-7 N, F, and X groups (Figure 4A). Then, according to IHC, PEDV was mainly located in the cytoplasm of enterocytes. In the positive control, PEDV was scattered in destroyed villous. In contrast, there was no PEDV detected in the negative control (Figure 4B). All tissues (duodenum, jejunum, and ileum) of CKT-7 T15, NF, F, and X groups had PEDV. The CKT-7 T15N strain was only identified in the jejunum. Furthermore, the CKT-7 N strain was dominant in the duodenum and ileum (Figure 4B). Overall, all strains proliferated in the small intestines, and although several strains were attenuated, they remained virulent.

### 3.7. Analysis of the Phylogenetic Relationship

After evaluating the pathogenicity of CKT-7 strains, their whole-genome sequences were analyzed using NGS to compare viral genome characteristics with other strains. The full-length genome was approximately 28 kb nucleotides long. The genetic diversity was demonstrated by a phylogenetic tree generated based on the nucleotide sequence of the whole genome, as well as ORF1a, ORF1b, S, ORF3, E, M, and N genes between the CKT-7 strains and reference strains. The phylogenetic trees were classified into genogroup 1 and 2, which were further divided into sub-genogroups 1a, 1b, and 2a, 2b, respectively. In general, genogroup 1 was considered classical with low pathogenicity, while genogroup 2 was regarded as field epizootic or panzootic with high pathogenicity, depending on the S gene. Forty reference strains were used to make the cluster, and the CKT-7 strains were assigned to genogroup G2-b. In the case of the whole genome, the CKT-7 strains belonged to genogroup 2 and closely clustered with the OH851 strain in an adjacent clade (Figure 5). The OH851 strain was first identified in June 2013 in the state of Ohio in the United States and is widely known as a low-virulent strain [17]. In ORF1a/b, the CKT-7 strains were organized by a novel clade, and the developed strains were composed of a sub-clade (Appendix A). In terms of S gene classification, the CKT-7 strains were categorized by a novel clade from genogroup 2, and the JSCZ1601 strain was closed in an adjacent clade (Appendix A). The JSCZ1601 strain was first reported from an outbreak in China in 2016, which had novel genetic characteristics, including a two-amino-acid deletion in the S protein and one amino-acid insertion in the 5′ UTR, and was revealed to be highly pathogenic to neonatal pigs [18]. Furthermore, there was no confirmed difference in phylogenetic relationships between the parental and developed strains according to the S gene. However, serial passage resulted in the phylogenetic diversity of each strain. In the group based on ORF3, M, and N, the CKT-7 strains were constructed by a novel clade from genogroup 2 (Appendix A). The CKT-7 strains were scattered to various groups based on the E (Appendix A). Altogether, the CKT-7 strains were included in a novel clade from genogroup 2, and the clade was composed based on the developed strains. Therefore, passage conditions altered genetic characteristics.

### 3.8. Comparison of Whole-Genome Sequence

To confirm genetic alterations during the serial passage, the whole-genome sequences of CKT-7 strains were compared. Among the CKT-7 strains, 119 mutations were observed, with 48 mutations in the non-structural protein region and 71 mutations in the structural protein region (Appendix A). The S region had the most mutations, with 56 observed. In ORF 1a/b region, the CKT-7 NF strain had the most mutations, with 25 mutations (A2, S3, N4, H5, V6L, T7H, L8W, A9L, A11K, N12R, D13P, A14S, E15Q, C278F, K939E, P981S, S1564F, D1573A, N1688D, I2710T, N3088K, N4127I, Y5097F, P5128L, T6415I) compared to the parental strain. On the contrary, the CKT-7 T15 strain had the fewest genetic mutations, with only six observed (P981S, T1531N, V3325F, A3545V, A4224S, M6455I). The CKT-7 N strain had 12 mutations in the ORF 1a/b gene region, including a deletion of one amino acid (aa) (F1457). Compared to the parental strain, the most common mutation observed in all CKT-7 strains was P981S. The S gene region had many mutations, with CKT-7 F having the most mutations (27) (N27A, V336 L, L337F, G612V, I662V, K773M, A828S, K892R, K983T, G965A, F966L, T967S, A969S, A970S, I1167S, I1223V, V1228A, N1273T, C1357F, F1380*, E1381, K1382, V1383, H1384, V1385, Q1386, *1387). The least variable strain was CKT-7 T15, which had only 16 aa mutations (N27K, T133P, A366S, I400F, S887R, D1253Y, I1350F, C1354F, F1380*, E1381, K1382, V1383, H1384, V1385, Q1386, *1387). Interestingly, an early stop codon was observed in the CKT-7 strains due to a single amino acid substitution (F1380*), which caused a 7 aa deletion in the C-terminus of the S protein. The C-terminus of the PEDV S protein contains two motifs involved in intracellular localization: the YxxΦ motif, which is a tyrosine-based motif (YEVF or YEAF) responsible for the intracellular localization of the S protein, and KVHVQ, which plays a role as the ER retrieval signal. Both motifs have been shown to be related to pathogenicity [19,20]. The early stop codon in CKT-7 strains indicates that mutations resulted in the truncated YxxΦ motif and deletion of the ER retrieval signal, KVHVQ. The remaining structural protein regions (ORF3, E, M, and N) showed random mutations. Notably, the CKT-7 N strain exhibited three mutations in the N region. Taken together, these findings suggest that genetic mutations occur depending on virus passage conditions, and these mutations can provide valuable information on the genomic characteristics, mutant information, pathogenic regions, and motifs. 

## 4. Discussion

Since its emergence in Asian countries, including the Republic of Korea, Japan, Taiwan, and China, PED outbreaks have been reported annually, resulting in significant economic losses for the swine livestock industry [21]. To prevent the spread of PED, the development of vaccines has been considered a valuable strategy. Live attenuated and inactivated vaccines are the most effective and traditional approaches because whole virus vaccines have sufficient antigenicity and immunogenicity to induce immunity [22]. However, the use of LAVs has a few drawbacks, such as the possibility of relapse to virulence, the need for high-titer virus cell adaptation, and difficulties in selecting highly immunogenic strains. In addition, inactivated vaccines require an inactivation process, which necessitates a multiple-dose vaccination program to increase immunogenicity. Until recently, genogroup 1 strains circulating in the Republic of Korea and several vaccines generated by the passage of classical PEDV strain (CV777, DR13) in cell culture systems have been developed. Nevertheless, the efficacy of these LAVs against recently emerged PEDV strains was disputed due to genetic mutations, indicating the necessity of developing a new vaccine against circulating virulent strains [23]. 

In this study, we investigated the role of proteases in enhancing the infectivity of viruses, including PEDV. Although the detailed mechanism of protease activity is not fully understood, it is known that proteolysis triggers the cleavage of viral GPs, facilitating their attachment to receptors and fusion with the host cell membrane. For example, the GP hemagglutinin of the influenza virus is cleaved by the cellular protease furin, while the S protein of murine hepatitis coronavirus (MHV) is processed by furin endo-protease [24,25]. In the case of the Ebola virus, the host protease cathepsin plays an essential role in GP-dependent entry by digesting viral GP 1 [26]. These viruses (influenza, MHV, and Ebola) have been shown to rely on host-derived proteases for enhanced infectivity. Conversely, PEDV is known to require proteases such as trypsin for isolation and efficient in vitro infection [27]. Previous reports have demonstrated that the role of trypsin, as a serine protease, is to catalyze the hydrolysis of the peptide bond at the carboxyl side of Arg and Lys [28]. The S protein of PEDV plays a crucial role in virus entry by being cleaved into two subunits, N-terminal S1, and C-terminal S2, by the exogenous protease trypsin. The S1 subunit attaches to the target receptor, while the S2 subunit fuses with the cell M [29]. Interestingly, a previous study has shown that trypsin-independent PEDV strains can be generated by adding other proteases to the cell culture media, such as elastase and bile acid. These findings suggest that bile acid may contribute to the high titer and low pathogenicity of PEDV strains [30]. Meanwhile, the metabolic role of bile acid in virus propagation is not well understood. Several viruses, such as porcine enteric calicivirus (PEC), have been found to utilize bile acid for their propagation, with PEC being able to adapt to bile acid-only culture systems [31]. In addition, bile acid has been shown to affect the infectivity of HCV, a causative agent of chronic liver disease [32]. Generally, bile acid is produced from cholesterol in the liver, stored in the gallbladder, and then secreted into the duodenum to support the absorption of fats by forming micelles [33]. 

Recent research has demonstrated a correlation between genetic alterations and pathogenicity, which has been proven by the reverse genetic system. Previous studies have revealed that the inactivating Endo U activity (aa substitution H226A of nsp 15) is related to the pathogenicity in seven-day-old-piglets [34]. Another study has suggested that the inactivation of the 2′-O-methlytransferase containing a catalytic tetrad (amino acid residues K, D, K, and E) and the endocytosis signal (Y1378A) of the S protein was attenuated when compared to the control [12]. In the case of the structural protein region, the deletion of 197 aa of S protein has been reported to reduce the mortality rate and the fecal virus shedding in four-day-old piglets [35]. CM Lin et al. have suggested that the PC22A-P120 strain has a premature stop codon of S protein due to the deletion of 9 aa (EVFEKVHVQ) which includes the partial YxxΦ (tyrosine-based motif YEVF or YEAF) and KVHVQ in the cytoplasmic tail. Although the mechanism of both motifs has not been fully understood, the YxxΦ motif is known as the endocytosis signal, and the KVHVQ motif is involved in the endoplasmic reticulum retrieval signal. The results showed that cell fusion activity increased in vitro, and virulence decreased in vivo [36]. In addition, the YxxΦ and KVHVQ motifs regulated the surface S protein level, and the deletion of both motifs had low pathogenicity in piglets that were confirmed to increase the villus height to crypt depth ratio and decrease the clinical symptoms and fecal viral shedding [20]. Similarly, the CKT-7 strains showed that the deletion of 6 aa (KVHVQ), an S protein retention signal, in the cytoplasmic tail at the S protein resulted in low virulence, including no mortality rate, less severe diarrhea rates, and mild intestinal lesions in piglets.

In this study, we isolated a virulent PEDV in the Republic of Korea and passaged it serially using various cell culture conditions up to 180 passages. We then analyzed the biological characteristics of developed strains, including morphological changes and growth kinetics. To evaluate the pathogenicity of the strains, we orally administered them to five-day-old pigs and monitored diarrhea and viral shedding. Subsequently, the whole-genome sequence of the strain was determined using NGS, and the genetic alterations were identified through comparative analysis. Our data revealed that the strains grown with TPCK-treated trypsin (T15, T15N) exhibited typical CPEs, such as syncytium and cell fusion, and also the titer was calculated to be approximately 7.0 log_10_TCID_50_/mL at 180 passages. Despite continuous passages, the pathogenicity of the strains remained. Although diarrhea and viral shedding in the piglets were postponed compared to the positive control, diarrhea, and viral genome in the rectal swab were still detected. In the case of the CKT-7 N, NF, F, and X strains, the strains were adapted to various cell culture conditions, but the CPEs were atypical, such as cell lysis and detachment. The CKT-7 N strain had the highest viral titer, peaking at approximately 8.67 ± 0.29 log_10_TCID_50_/mL, but it caused no pathogenicity, including mortality and symptoms, in the piglets. The CKT-7 NF, F, and X strains exhibited typical CPEs, such as cell lysis, cell fusion, and cell detachment, while still retaining pathogenicity and low titers, making them unsuitable for use as LAV candidates. This study’s results suggest that an effective LAV against PEDV may need a suitable protease such as bile acid. Overall, the CKT-7 N strain had low pathogenicity and showed potential as one of the LAV candidates.

Further studies will be conducted to evaluate the efficacy of vaccines in sows. This is because vaccinated sows can develop immunity against PEDV, and their maternal antibodies are transferred to protect neonatal piglets through colostrum and milk. As the placenta of sows is impermeable, it is crucial to induce secretory immunoglobulin A (sIgA) in colostrum, especially during the neonatal period, to provide local passive immunity [37]. Therefore, we will assess whether vaccinated sows induce PEDV-specific sIgA in colostrum and whether the immunity induced can protect against virulent PEDV challenges. Additionally, reverse genetics technology can be used to analyze the changes in viral infectivity, pathogenicity, and functional domains in response to genetic alterations. 

## Figures and Tables

**Figure 1 vaccines-11-00923-f001:**
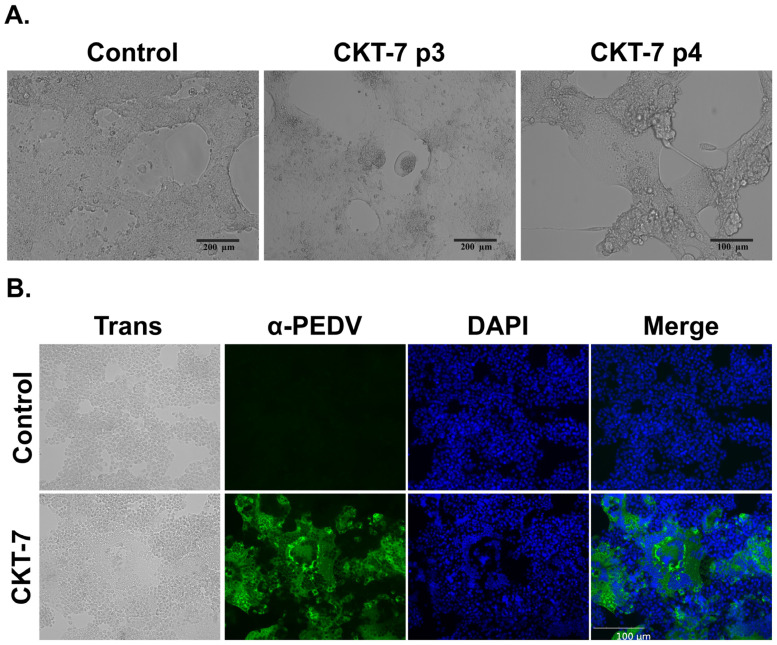
Isolation of CKT-7. (**A**) The development of cytopathic effects (CPEs) in Vero cells infected with CKT-7 (passage number 3 or 4) or uninfected control cells was observed using microscopy. (**B**) The identification of CKT-7 was confirmed through immunofluorescence assay using a porcine epidemic diarrhea virus (PEDV)-specific monoclonal antibody. Infected cells were fixed (first panel), incubated with anti-PEDV Mab (second panel), stained for the nucleus with 4′6-diamino-2-phenylindole (DAPI) (third panel), and merged (fourth panel).

**Figure 2 vaccines-11-00923-f002:**
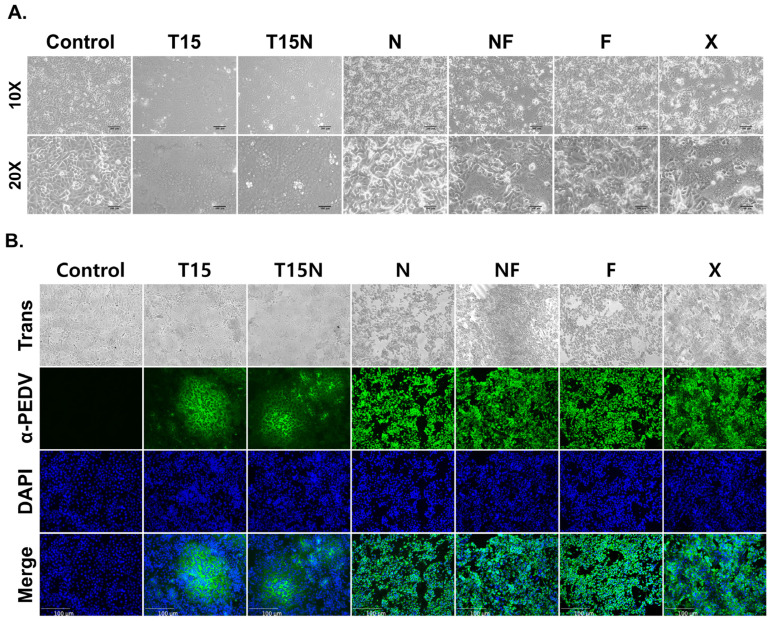
Observation of cytopathological and growth characteristics of CKT-7 strains. (**A**) Observation of the cytopathic effect infected with CKT-7 strains passage number 100 using microscopy at a magnification of 10× (top panels) and 20× (bottom panels). (**B**) An immunofluorescence assay was performed using PEDV-specific monoclonal antibody against the nucleocapsid (N) protein in the PEDV-infected Vero cell.

**Figure 3 vaccines-11-00923-f003:**
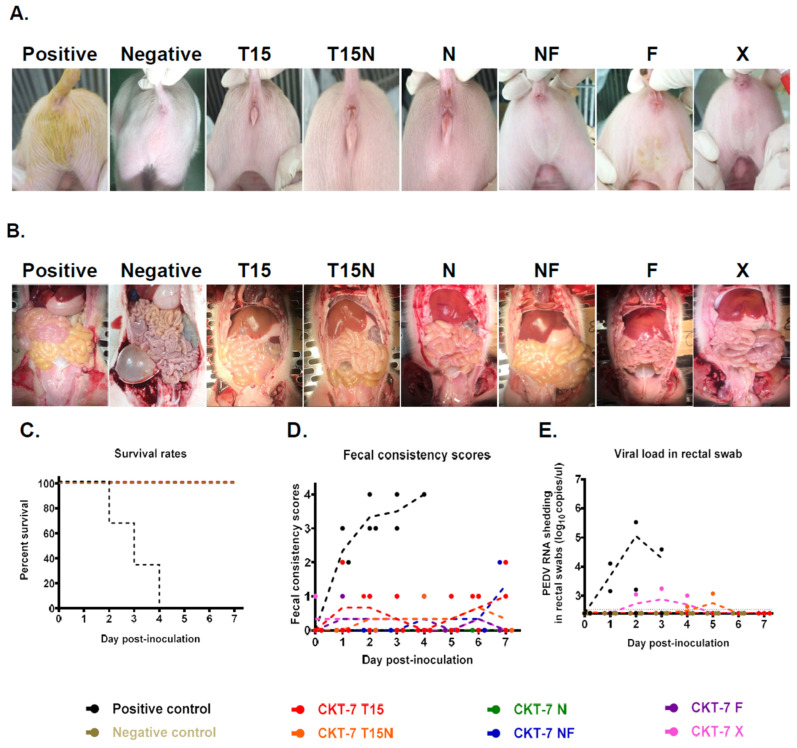
Evaluation of virulence of CKT-7 strains. (**A**) Clinical symptoms of piglets infected with PEDV passaged using six different culture conditions. (**B**) Gross lesions in the small intestines compared among the different groups after autopsy. (**C**) The survival rates of piglets inoculated with the CKT-7 strains. (**D**) Fecal consistency scores were measured according to the protocol described in the Materials and Methods. (**E**) Viral RNA shedding in the rectal swab was analyzed using quantitative real-time RT-PCR throughout the animal experiment. Each dot (.) represents an individual sample.

**Figure 4 vaccines-11-00923-f004:**
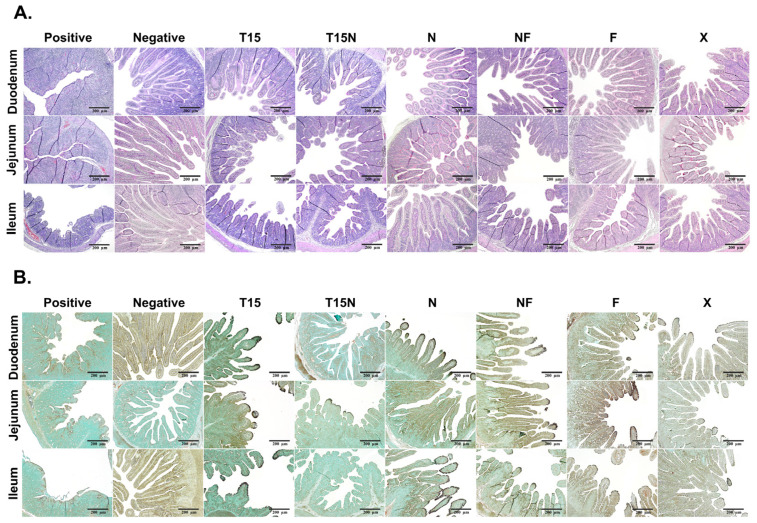
Analysis of histopathology and immunohistochemistry (IHC) in pigs inoculated with the CKT-7 strains. After autopsy, the small intestines from the pigs were classified into duodenum, jejunum, and ileum. The tissues were analyzed for pathogenicity by staining with Hematoxylin and eosin (H&E) and IHC using a PEDV-specific monoclonal antibody to detect antigens. (**A**) Tissues infected with CKT-7 strains stained with H&E. (**B**) IHC for detection of antigens using PEDV monoclonal antibody. The PEDV antigens were indicated by brown staining and were mainly observed in villous epithelial cells. Images of H&E and IHC staining images were taken at 100× magnification.

**Figure 5 vaccines-11-00923-f005:**
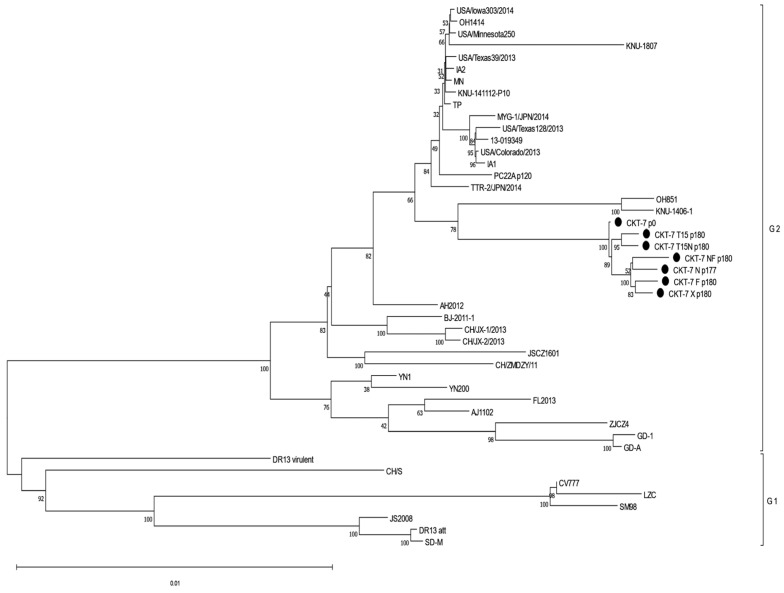
Analysis of phylogenetic relationship based on the nucleotide sequence of the whole genome. The trees were determined using the neighbor-joining method of MEGA X software (version 11). The bootstrap analysis was generated using 1000 replicates.

**Table 1 vaccines-11-00923-t001:** Generation of CKT-7 strains according to different culture conditions.

Strains	Passage Number																																		
Origin	1	2	3	…****	9	10	11	…	32	33	34	…	39	40	41	42	…	51	52	53	54	…	60	61	62	63	64	65	66	…	70	71	72	…	180
T15 *	1	2	3	…	9	10	11	…	32	33	34	…	39	40	41	42	…	51	52	53	54	…	60	61	62	63	64	65	66	…	70	71	72	…	180
T15N **					1	2	3	…	24	25	26	…	31	32	33	34	…	43	44	45	46	…	52	53	54	55	56	57	58	…	62	63	64	…	172
T10N									1	2	3	…	8																						
T5N														1	2	3	…	12																	
T2N																			1	2	3	…	9												
N																								1	2	3	4	5	6	…	10	11	12	…	120
NF ***																											1	2	3	…	7	8	9	…	117
X																															1	2	3	…	111
F																																1	2	…	110

* TPCK-treated trypsin (T): L-1-tosylamide-2-phenylethyl chloromethyl ketone-treated trypsin; ** Na-GCDCA (N): sodium glychenodeoxychloic acid; *** FBS (F): fetal bovine serum, **** Omitting numbers.

**Table 2 vaccines-11-00923-t002:** Titer of the CKT-7 strains according to passage number.

T15		T15N		N		NF		F		X	
Passage No.	Titer *	Passage No.	Titer	Passage No.	Titer	Passage No.	Titer	Passage No.	Titer	Passage No.	Titer
P10	3.50 ± 0.00	P10	3.67 ± 0.14	P10		P10		P10		P10	
P20	4.58 ± 0.14	P20	4.75 ± 0.00	P20		P20		P20		P20	
P30	5.42 ± 0.29	P30	5.92 ± 0.14	P30		P30		P30		P30	
P40	6.00 ± 0.25	P40	5.42 ± 0.14	P40		P40		P40		P40	
P50	6.17 ± 0.14	P50	6.50 ± 0.25	P50		P50		P50		P50	
P60	5.33 ± 0.29	P60	5.33 ± 0.52	P60		P60		P60		P60	
P70	6.08 ± 0.38	P70	5.50 ± 0.00	P70		P70		P70		P70	
P80	6.00 ± 0.00	P80	5.85 ± 0.14	P80	7.17 ± 0.14	P80	6.67 ± 0.14	P80	6.33 ± 0.14	P80	6.75 ± 0.00
P90	6.33 ± 0.14	P90	5.75 ± 0.43	P90	7.08 ± 0.29	P90	7.00 ± 0.43	P90	6.83 ± 0.38	P90	6.92 ± 0.38
P100	5.50 ± 0.00	P100	4.83 ± 0.14	P100	7.25 ± 0.43	P100	5.92 ± 0.14	P100	6.83 ± 0.14	P100	6.75 ± 0.25
P110	6.58 ± 0.14	P110	7.08 ± 0.14	P110	7.42 ± 0.38	P110	6.67 ± 0.52	P110	6.75 ± 0.00	P110	6.42 ± 0.29
P120	6.33 ± 0.14	P120	6.92 ± 0.38	P120	7.83 ± 0.29	P120	7.83 ± 0.29	P120	7.00 ± 0.00	P120	7.17 ± 0.52
P130	6.58 ± 0.14	P130	6.42 ± 0.29	P130	8.25 ± 0.25	P130	7.08 ± 0.38	P130	7.08 ± 0.29	P130	6.83 ± 0.38
P140	7.42 ± 0.14	P140	7.67 ± 0.29	P140	8.67 ± 0.29	P140	7.28 ± 0.29	P140	7.33 ± 0.14	P140	7.25 ± 0.25
P150	8.30 ± 0.14	P150	8.50 ± 0.00	P150	8.50 ± 0.25	P150	7.08 ± 0.14	P150	7.17 ± 0.14	P150	7.42 ± 0.52
P160	7.33 ± 0.38	P160	7.08 ± 0.29	P160	8.25 ± 0.66	P160	8.33 ± 0.14	P160	7.92 ± 0.14	P160	7.83 ± 0.14
P170	7.25 ± 0.25	P170	7.50 ± 0.25	P170	8.00 ± 0.00	P170	7.67 ± 0.14	P170	7.00 ± 0.25	P170	6.83 ± 0.29
P180	7.00 ± 0.00	P180	7.83 ± 0.14	P180	8.25 ± 0.25	P180	7.92 ± 0.14	P180	7.25 ± 0.25	P180	6.92 ± 0.14

* log_10_TCID_50_/mL.

**Table 3 vaccines-11-00923-t003:** Summary of virulent studies of CKT-7 strains.

Group	Inoculum	Route	No. of Pigs	Mortality Rate * [% (No/Total)]	Severe Diarrhea Rate [% (No/Total)]	Clinical Symptoms	Virus Shedding	Peak Fecal Virus Shedding Titer [log10 Copies/μL], dpi
1	Positive control	Oral	3	100 (3/3)	100 (3/3)	NA ** (>4)	Started at 1 dpi	5.05 ± 0.24, 2
2	Negative control		3	0 (0/3)	0 (0/3)	No diarrhea	N/D ***	NA
3	CKT-7 T15		3	0 (0/3)	0 (0/3)	Mild	Limited	NA
4	CKT-7 T15N		3	0 (0/3)	0 (0/3)	Mild	Started at 4 dpi	2.48 ± 0.13, 5
5	CKT-7 N		3	0 (0/3)	0 (0/3)	No diarrhea	N/D	NA
6	CKT-7 NF		3	0 (0/3)	0 (0/3)	Mild	Limited	NA
7	CKT-7 F		3	0 (0/3)	0 (0/3)	No diarrhea	N/D	NA
8	CKT-7 X		3	0 (0/3)	0 (0/3)	No diarrhea	Started at 2 dpi	2.68 ± 0.49, 3

* Fecal consistency scores: 0, solid; 1, pasty; 2, semi-liquid; 3, liquid; 4, death. Fc score of 3 was regarded as severe diarrhea; ** not available; *** non-detection.

## Data Availability

Not applicable.

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
