# Peer review of "Development of Effective PEDV Vaccine Candidates Based on Viral Culture and Protease Activity"

_vaccines, 2023, doi:10.3390/vaccines11050923_

Round 1

Reviewer 1 Report

This is a relatively high quality research on development of PEDV live attenuated vaccine (LAV) using the traditional passaging method. They tried several treating methods during the passaging the viruses and one strain is a possible candidate for developing vaccine. The article was organized well and the research was solid. However, major and minor revisions of the article are needed before publication.

Major revisions:

1. The present title is more like one for a review article, therefore, a new title is suggested.

2. More discussion on methods is needed, e.g. what's the principle for the strategies of TPCK and GCDCA supplementation? Discussion of GCDCA role is not enough.

3. The article can be revised to be more concise. For instance, in 3.5 the pathogenecity evaluation, it is only necessary to talk the main results or findings which help the reader to get the main point the authors want to present. 

Minor revisions:

1.material and method 2.6. Is it enough to provide the life support for the newborn piglets to feed them replacer three times per day? Please provide the manufacturer of the replacer.

2. Table 4 can be removed or as supplemental information. In Figure 5, genetic trees of structural and nonstructural genes can be removed except the analysis for the whole genome. 

Many expressions in the article need to be improved or corrected. For instance:

1.Line199, author used "prominent CPE" and "major CPE", what's the difference between the both? maybe "obvious" is more proper here than "prominent".

2. line 329, the intestinal tissues of pigs were classified into duodenum, jejunum and ileum after autopsy. The author probably wanted to say that the small intestine was generally divided into three parts, duodenum, jejunum and ileum.

3. line416, KVHVQ plays a role in ER retrieval signal, here "plays role as" could be better.

A major language revision is suggested.

Reviewer 2 Report

Known in the field based on previous literatures:

1. Porcine epidemic diarrhea (PED) is caused by a coronavirus named porcine epidemic disease virus (PEDV). Virus infects the cells lining the small intestine of a pig and causes porcine epidemic diarrhea.

2. Porcine epidemic diarrhea virus (PEDV) causes acute diarrhea, vomiting, dehydration and high mortality in neonatal piglets.

3. Effective vaccines for PEDV were recently developed or are still in development.

In this article authors reported following findings:

I have gone through the article titled "Strategies for selecting PEDV vaccine candidates based on viral culture and protease activity’. Authors isolated a virulent Korean strain, CKT-7, from a piglet with severe diarrhea and employed different condition and serial passages of the strain to get live attenuated vaccine (LAV) candidates. Authors performed and reported following findings-

1. Authors isolated PEDV from porcine intestines and generated live attenuated vaccine candidates.

2. Authors studied and analyzed the biological characteristics of developed strains including morphological changes and growth kinetics.

3. The results suggested that LAV candidates can be generated through serial passage with different culture conditions.

The article presented are interesting and generally supportive of the conclusions drawn. The following minor suggestions if incorporated could help in the better understanding of the significance of the work and implications.

Minor Concerns:

1. As authors used different condition and several number of passages to get LAV, at which passages CKT-7 strain started to show weak pathogenicity?

2. Authors should also include scale bar in figure 2A. Is there any reason to escape the scale bar?

3. There are different kind of proteases are available and their functions vary as discussed by authors too. What was the criteria for selection of proteases used in your study? Please discuss it.

4. Several studies are already available about derivation of attenuated porcine epidemic diarrhea virus as vaccine candidate. Explain, how your study is different from rest and how does it address a specific gap in the field?
